# Potential Functions of the Gastrointestinal Microbiome Inhabiting the Length of the Rat Digest Tract

**DOI:** 10.3390/ijms20051232

**Published:** 2019-03-12

**Authors:** Dongyao Li, Haiqin Chen, Jianxin Zhao, Hao Zhang, Wei Chen

**Affiliations:** 1State Key Laboratory of Food Science and Technology, Jiangnan University, Wuxi 214122, China; 7130112040@vip.jiangnan.edu.cn (D.L.); jxzhao@jiangnan.edu.cn (J.Z.); zhanghao@jiangnan.edu.cn (H.Z.); weichen@jiangnan.edu.cn (W.C.); 2School of Food Science and Technology, Jiangnan University, Wuxi 214122, China; 3National Engineering Research Center for Functional Food, Jiangnan University, Wuxi 214122, China; 4Beijing Innovation Center of Food Nutrition and Human Health, Beijing Technology and Business University (BTBU), Beijing 100048, China

**Keywords:** rat, gastrointestinal tract, microbiome function, ecosystem service

## Abstract

The rat is an important model animal used frequently in biological researches exploring the correlations between gut microbiome and a wide array of diseases. In this study, we used an extended ancestral-state reconstruction algorithm to predict the functional capabilities of the rat gastrointestinal microbiome. Our results indicate an apparent tendency toward metabolic heterogeneity along the longitudinal and transverse axes of the rat gastrointestinal tract (GIT). This heterogeneity was suggested by the enriched small-molecule transport activity and amino acid metabolism in the upper GIT, the aerobic energy metabolism in the stomach and the mucolysis-related metabolism in the lower GIT mucus layer. In contrast to prior results, many functional overlaps were observed when the gastrointestinal microbiomes of different hosts were compared. These overlaps implied that although both the biogeographic location and host genotype were prominent driving forces in shaping the gastrointestinal microbiota, the microbiome functions were similar across hosts when observed under similar physicochemical conditions at identical anatomical sites. Our work effectively complements the rat microbial biogeography dataset we released in 2017 and, thus, contributes to a better understanding and prediction of disease-related alterations in microbial community function.

## 1. Introduction

Symbiotic microorganisms located inside and on the surface of the host body provide many biological functions that are not encoded in their host genome [1]. The complementary functions are generally considered as ecosystem services which are crucial to maintaining the host physiological homeostasis [2]. In recent years the gut flora has attracted increasing attention in several fields and is, now, even considered a hidden organ [3], as its collective metabolic activity is somewhat equal to a virtual organ and profoundly influences the host health and disease in both direct [4,5,6] and indirect manners [7,8,9,10].

In the past decade, encouraged by the progress in high-throughput sequencing technology and new developments in bioinformatics, the scientific community has continuously and gradually investigated the gut microbial communities underlying biological functions observed in human- [11,12] and host-associated [13,14] habitats. In 2008, the National Institutes of Health (NIH) initiated the Human Microbiome Project (HMP), which aimed to characterise the microbiota within 5 main human body habitats (i.e., airway, skin, oral, gut and urogenital) from the aspects of community structure and metabolic potential [15]. In 2010, the MetaHIT project released the first human gut microbial gene catalogue established by metagenomic sequencing [16]. In 2015, the Beijing Genomics Institute (BGI) released the first mouse gut microbial gene catalogue [17]. These catalogues have since served as valuable references for numerous studies. However, most of these studies used functional gene sets derived from faecal samples, although the appropriateness of using faeces as a proxy for the whole gastrointestinal tract (GIT) remains unconfirmed.

In 2017, our research group released the first microbial biogeography of the rat GIT, which systematically characterised the baseline microbial structure, membership, diversity and ecology in this long-standing biomedical research animal model [18]. By comparing different niches along the length of the rat digestive tract, we concluded that faecal samples could not fully represent the microbiota throughout this tract, at least, in terms of taxonomic profiling. In an ecosystem such as the GIT, the collective function is co-determined by the population composition and the activities of each component. The potential metabolic activities of a microbe are encoded by its genome, and many host-associated microorganisms, even those only distantly related, may share functional genes and, thus, thrive in the hash gastrointestinal (GI) environment [16]. The HMP also indicated that the community function remained relatively stable, despite dramatic variations in the community structure [19].

Therefore, in this study, we took the next step and predicted the functional capabilities of the rat GI microbiome using an extended ancestral-state reconstruction algorithm [20] and projected these functions onto a Kyoto Encyclopaedia of Genes and Genomes (KEGG) orthology and pathway framework [21] to address the following questions: (i) What complementary ecosystem service functions do microbial residents provide along the length of the rat digestive tract? (ii) Does the faecal functional profile fully represent those of other GI segments? (iii) To what extent dose heterogeneity in the community composition reflect the differences in the community function across biogeographic locations and hosts?

## 2. Results and Discussion

The rat genome was published in 2004 and was the third mammalian genome sequenced [22] after the human [23] and mouse genomes [24]. However, the rat intestinal microbiome, as its second genome, was not comprehensively characterised until 2017. Our previously published rat microbial biogeography filled this gap and suggested that the stratification of microbial community and the shift in microbial metabolites might interact, serving as both cause and effect [18].

### 2.1. Metabolic Heterogeneity in the Community Function

In present study, we aimed to characterise the rat GI microbiome from the perspective of a functional potential. Based on a total of 6009 KEGG orthologue (KO) annotations, we observed an apparent tendency toward a metabolic heterogeneity along the longitudinal and transverse axes of the rat GIT. This tendency was intuitively displayed on the principal component analysis (PCA) ordination plots (Figure 1A–C), where samples were grouped by different biogeographic factors such as the anatomic region (adonis: *R*^2^ = 0.47; *p* ≤ 0.001), subsite (adonis: *R*^2^ = 0.60; *p* ≤ 0.001) and niche location (adonis: *R*^2^ = 0.24; *p* ≤ 0.001). The first principal component (PC1) explained 53.75% of the total heterogeneity and parsed out the niche locations of the samples, while the second principal component (PC2) explained 18.8% and mainly parsed out the anatomic regions.

The KOs acting as the strongest drivers of this trend in heterogeneity are plotted as loadings on the PCA plot, with the arrow lengths proportional to their contributions (Figure 1A). In the second quadrant, 4 KOs point to the colonic mucosal samples, indicating that these KOs play significant roles in the local agglomeration of these samples. Here, 3 KOs, K02025, K02026 and K02027, constituted the elements of a putative multiple sugar transport system, while the fourth, K03088, was an RNA polymerase factor. Similarly, K02025, K02026 and K03088 are also enriched in the mouse gut microbial gene catalogue and drive the separation of the human and mouse gut microbiomes [17]. By partitioning the metagenome functional contributions, we found that the major contributing operational taxonomic unit (OTU) to this ATP-binding cassette (ABC) transporter belonged to the Clostridium cluster XIVa, in which many species produce butyrate and specifically colonise the mucins close to the intestinal epithelium [25].

In the first quadrant, 6 KOs point to the small-intestinal samples. Three of these KOs, K03293, K03294 and K08659, might be involved in the catabolism of free small peptides and amino acids from the upper GIT digestive residue [26]. Interestingly, the corresponding contributing OTUs all belonged to Lactobacillus, although the OTU ranks varied between different KOs (Figure 1D). Studies on gut microbiota have rarely reported the proteolytic activity of Lactobacilli in GIT. However, Lactobacillus strains have occasionally been used as starter cultures to eliminate gluten toxicity [27] and to ferment meat products [28] in the field of food microbiology, suggesting that Lactobacillus species may play specific roles in rat small-intestinal habitats. Based on our prior phylogenetic prediction result, the molecular species participating in this process were probably *Lactobacillus intestinalis*, *L. hominis*, *L. johnsonii*, *L. taiwanensis*, *L. gasseri* and *L. saniviri* [18]. Additionally, OTU214919, which was previously annotated as Turicibacter, comprised a major contributory phylotype in the orthologous gene family K07024, with a much higher degree of intersubject variation regarding contributions (note: Turicibacter is another predominant genus throughout the rat digestive tract).

However, the permutational multivariate analysis of variance (PERMANOVA) result revealed that the individual subject was no longer a significant categorical factor (adonis: *R*^2^ = 0.09; *p* = 0.18) with respect to community function, in contrast to the situations regarding structure and membership [18]. To some extent, this weakened stratification was in line with the HMP results, which showed the taxonomic variety versus metabolic stability within a healthy population [19].

### 2.2. Module-Centric Metabolic Reconstruction of the Rat Gastrointestinal (GI) Microbiome

A KEGG pathway module is a collection of manually defined, tight functional units which allow a higher-level grouping of KO gene families into pathways or functional classifications [29]. To summarise the community function of the rat GIT at a higher level, 6009 KOs were collapsed into 125 metabolic modules via a parsimony approach. The abundance varied significantly in at least one sampling site in more than half (75/125, 60%) of these detected modules, demonstrating the uniqueness of various GI niches (Figure 2A and Appendix A). Generally speaking, the structural complexes that processed the environmental information were prevalent in the upper GIT. These complexes were principally transporters of small molecules (saccharides: M00194, M00197, M00201, M00207, M00276 and M00277; polyols: M00200; phosphates: M00222; amino acids: M00230, M00232, M00235 and M00237; peptides: M00333, M00348 and M00349; mineral and organic ions: M00185, M00193, M00299, M00300 and M00319). This phenomenon suggests an active exchange of materials and information between commensal microbiota and the digestive milieu, wherein the macromolecules in foodstuffs are broken down, leading to a relative abundance of free small molecules [30].

Furthermore, the majority of amino acid metabolic modules (M00016, M00017, M00022, M00033, M00034, M00035, M00036 and M00045) were also prevalent in the upper GIT. In addition to the bacterial proteasome module M00342, this prevalence might be associated with the ordination result of PCA (Figure 1A). However, the module M00018, which comprises serine and threonine metabolism, was abundant in the caecal mucus layer, likely because active bacterial mucolysis within this habitat involves the breakdown of serine- and threonine-rich O-linked mucins [31]. The hydroxyl groups of these amino acids are always linked to various uronic acid-rich oligosaccharide side-chains [32], and the corresponding metabolic module M00061 was also detected abundantly in the proximal-colonic mucus layer.

However, in some cases, our attempts to tile a whole pathway suggested the simultaneous existence of alternative modules with distinct internal structures but identical outlines enabling them to fill the same metabolic gaps. In the rat GI microbiome, we identified several such biological situations. For example, either the mevalonate (M00095) or non-mevalonate (M00096) module could perform terpenoid backbone biosynthesis; the former and the latter were enriched in the upper (i.e., jejunal contents) and lower GIT (i.e., proximal-colonic contents), respectively. Similar phenomena were also observed for the modules M00157 (bacterial F-type ATPase; enriched in jejunal contents) and M00159 (prokaryotic V-type ATPase; enriched in middle-colonic contents). These results suggest that when facing various survival challenges, the microbiota inhabiting different GI niches might use distinct strategies to complete the pathway.

We previously found that the rat core microbiota evolved from an aerobic to facultative anaerobic to obligatory anaerobic metabolism along the longitudinal axis [18], in accordance with imaging observations of the oxygen gradient throughout the GIT [33]. This tendency was also evident when biomarkers of community function were mined at different sampling sites; here, functional modules involved in aerobic heterotrophic metabolism (M00008, M00009, M00011, M00012, M00115, M00144 and M00149) were prevalent in the stomach. Besides the longitudinal axis, in the radial direction, an intraluminal oxygen gradient further extends radially from the mucosal tissue interface into the lumen [34]. In the rat lower GIT, both the mucus layer and lumen are densely populated by various microorganisms, which separately form topologically distinct co-occurrence networks [18]. To compare the functions of these closely adjacent compartments, the linear discriminant analysis (LDA) effect size (LEfSe) system was applied only to mucosal and luminal samples from the large intestine, limiting pairwise comparisons performed between the same subsites. The results demonstrated a dramatic abundance variance in approximately a third (39/121, 32.2%) of the modules detected in these samples (Figure 2B and Appendix A). Specifically, modules associated with aerobic metabolism (M00009, M00011, M00144, M00149, M00156 and M00159) were over-enriched in the mucus layer, whereas those associated with anaerobic metabolism (M00001 and M00002) were under-enriched.

The GI microbiome is an important supplier of vitamins [35]. Here, we observed elaborate supply patterns in different biogeographic niches. Modules associated with the biosynthesis of vitamins B_2_, B_3_ and B_12_ (M00125, M00115 and M00122, respectively) were prevalent in the upper GIT, whereas those associated with the biosynthesis of the vitamins B_1_, B_5_, B_7_ and B_9_ (M00127, M00119, M00123 and M00126, respectively) were prevalent in the lower GIT. Particularly, a pairwise comparison of the microbial metagenomes in the mucus layer and lumen revealed a high abundance of the pyridoxal biosynthesis module M00124 (vitamin B_6_; cofactor for many enzymes with amino acid substrates [36]) in the former, suggesting ubiquitous proteolytic activities in this niche location.

### 2.3. Functional Overlaps in the Murine GI Microbiome

Previously, we compared the GIT microbiota of different rodents and concluded that the rat biogeographic map might serve as a new reference for digestive tract-related disease research [18]. In this study, we reanalysed the datasets [37,38,39,40,41] using the identical computational steps applied to rats to further investigate the GIT microbial metabolic similarities and differences among various mammalian hosts. Unsurprisingly, the human samples had the lowest nearest sequenced taxon index (NSTI; mean = 0.094 ± 0.088 (standard deviation)), followed by the rat (0.122 ± 0.039), mouse (0.213 ± 0.065) and woodrat samples (0.218 ± 0.05).

When these samples were mapped on the same PCA plane, a lot of overlaps were observed among different murine hosts (Figure 3A) in contrast to the prior ordination result calculated from the phylotype abundances [18]. This observation was also confirmed when the ecological distances between samples from different hosts were calculated. Here, the Euclidean distances calculated from the OTU abundances were significantly higher than those calculated from the KO profiles (nonparametric paired Wilcoxon-signed-rank test, *p* < 2.2 × 10^−16^). Furthermore, regarding the functional overlap, human GI microbiome was more similar to the rat microbiome than the mouse microbiome within the large-intestinal mucus layer (5405 overlapping KOs between humans and rats versus 4583 overlapping KOs between humans and mice). The pattern of overlap resembled the pattern observed when the human and mouse gut catalogues were compared at the KO level (4,969 overlapping KOs between humans and mice) [17].

Besides the host genotype (adonis: *R*^2^ = 0.27; *p* ≤ 0.001), the anatomic region was also identified as a relatively weak categorical factor (*R*^2^ = 0.09; *p* ≤ 0.001) and was mainly parsed out during PC1 (56.71% variation explained). Coincidently, some of the loading KOs (K02025, K02026, K02027, K03088 and K01834) in rats also appeared on the PCA plot when 2 other rodents were considered (Figure 3B). In the third quadrant, K02025, K02026, K02027, K06147 and K09687, which comprise an ABC transporter, again pointed to the large-intestinal samples, while K01834 still pointed toward the small-intestinal samples in the fourth quadrant. Moreover, K12373, which corresponds to a beta-hexosaminidase involved in keratan sulfate degradation, fell into the second quadrant, suggesting that this orthologous gene family plays a major role in the bacterial mucolysis specific to the mouse large-intestinal microbiome.

## 3. Materials and Methods

The raw reads of 75 samples sequenced during our prior project, PRJNA324666 [18], were downloaded from the NCBI Short Read Archive (SRA). The quality control and operational taxonomic unit (OTU) picking were performed as previously described [18], whereas *de novo* OTUs were removed by keeping only those with matching Greengenes IDs [42]. Chimeras (identified using Broad Institute Chimera Slayer [43]) and singletons were also removed from the OTU table. During OTU picking, 84.23% of reads were mapped to references at a similarity level of 97%. Given the distances between the sequencing reads and reference seeds, however, the cluster representative names were permuted 10 times (with replacements) within the OTU table to avoid introducing biases when estimating the gene families present in microorganisms from the Phylogenetic Investigation of Communities by Reconstruction of Unobserved States’ (PICRUSt) pre-calculated files [20]. Next, the PICRUSt pipeline (version 1.1.3) was used to predict the sample-wide KEGG orthologue (KO) profiles based on the 10 permuted tables and to estimate the contributions of each OTU to the given KOs. The average correlation between the permutations within a sample exceeded 0.99 (Spearman’s rho). Hence, we calculated the arithmetically averaged abundances of KO groups per sample and used these values in all the downstream analyses. We further retrieved and processed raw reads from 5 other projects (mice: PRJNA178786 (35 samples retrieved) [37], PRJEB2233 (2 samples) [38] and figshare1499145 (85 samples) [39]; woodrats: PRJNA197212 (30 samples) [40]; and human: mgp1982 (17 samples) [41]) as described previously and above and combined these data with the KO profiles of rat samples.

The HUMAnN pipeline (HMP Unified Metabolic Analysis Network; version 0.99) [44] was selected to reconstruct the metabolic modules and thus collapse the predicted functions into a higher level. The parameter settings and processing modules within this pipeline were optimised for the best overall performance as previously described [44]; here, a parsimony approach [45] was used to conservatively assign gene sets to hierarchical metabolic modules. To normalise the sequencing depth, we used total-sum scalings to the KO profile and module abundance matrices and thus formed the substrate for the following statistical analyses.

Functional similarities between the samples were measured using Euclidean distances based on the KO profile matrix. PERMANOVA was used to statistically describe the relationships between the metabolic variability and sample metadata, and the significance of these relationships was tested using 999 Monte Carlo permutations. PCA was performed to map the KO profiles in a multidimensional space to a 2-dimensional plane. The calculation of the Euclidean distance, PERMANOVA and PCA were all performed using the R-package VEGAN (version 2.5-2) [46]. The PCA results were visualised using EMPeror [47], and both sample scores and KO loadings were plotted in the same coordinates. The LEfSe biomarker discovery tool (LDA Effect Size; version 1.0) [48] was applied to the module abundances from HUMAnN to determine the over- or under-enriched metabolic modules in at least one sampling site and in different niche locations; here, the alpha value of 0.01 and logarithmic LDA score threshold of 2.0 were applied. The metabolic modules were hierarchically organised in a dendrogram according to the KEGG BRITE database and visualised using GraPhlAn (Graphical Phylogenetic Analysis; version 1.1.3) [49]. Venn diagrams describing the function overlaps between hosts were constructed using the R-package VennDiagram (version 1.6.20 https://CRAN.R-project.org/package=VennDiagram).

## 4. Conclusions

In summary, we comprehensively characterised the putative biological functions of the rat GI microbiome, which complemented our previously released microbial biogeography project [18]. The rat digestive system composes many different niches with distinct physicochemical conditions and immune responses. We observed a number of specific functional clades ubiquitous within and characteristic to each of these niches. This spatially functional heterogeneity was consonant with our prior phylotype-based analysis, which revealed the tendency of GIT microbiota to form a stratified community structure and membership [18]. From the perspective of community function, we thus further confirmed that the faecal gene set cannot represent the full ecosystem service repertoire provided by the entire rat GIT microbiome. Furthermore, a comparison of the GI microbial metagenomes from different hosts revealed many functional overlaps. This phenomenon implies that although the GI microbiota is shaped significantly by the host genotypic background, many microbiome functions are shared across hosts, consistent with the similar physicochemical conditions at identical anatomical sites.

## Figures and Tables

**Figure 1 ijms-20-01232-f001:**
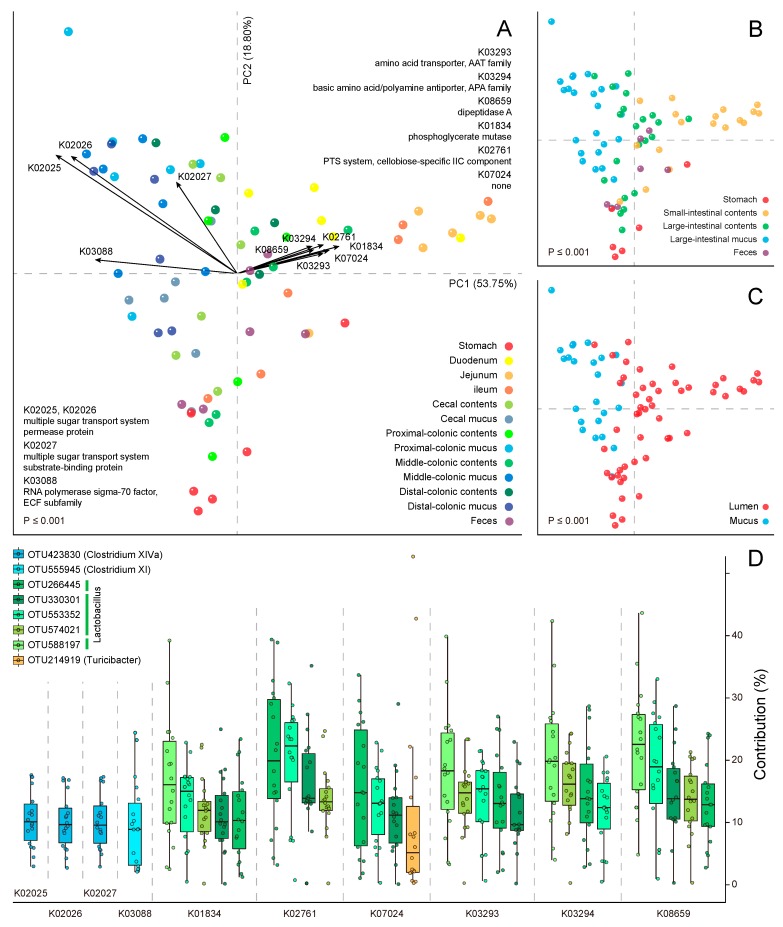
The spatial heterogeneity in the rat microbial community functions: a principal component analysis (PCA) based on the Kyoto Encyclopaedia of Genes and Genomes (KEGG) orthologue (KO) profiles of gut microbiome across the rat gastrointestinal tract (GIT), where potential relationships between metabolic variability and sample metadata are highlighted in panels **A** to **C** by different colorations. The permutational multivariate analysis of variance (PERMANOVA) results reveal the grouping of samples from the same (**A**) sampling site (adonis: *R*^2^ = 0.60), (**B**) anatomic region (*R*^2^ = 0.47) and (**C**) niche location (*R*^2^ = 0.24). The *p*-value of the Monte Carlo permutation test is shown on the lower left. The percentages of variation explained by first principal component (PC1) and second principal component (PC2) are listed along the axes representing them. The top 10 KOs driving the heterogeneity are indicated in panel **A**. For different anatomic regions, the major contributing operational taxonomic units (OTUs) and their contributions to these KOs are shown in panel (**D**).

**Figure 2 ijms-20-01232-f002:**
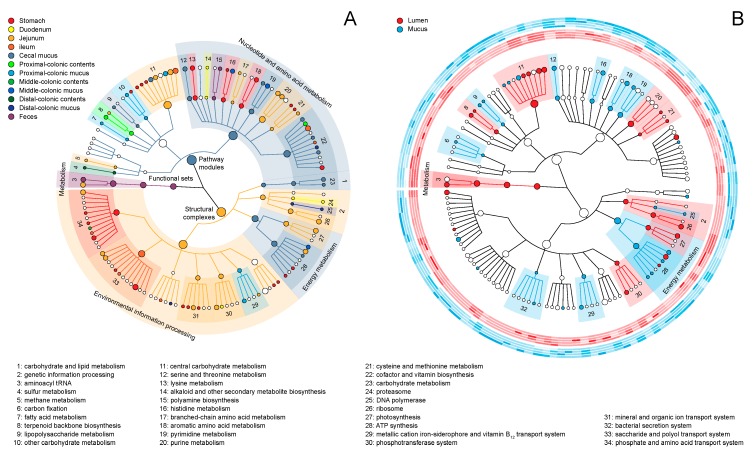
The modular difference in metabolic landscapes across the sampling sites and niche locations: The cladograms illustrate the differentially abundant metabolic modules under the KEGG BRITE hierarchy, determined using the linear discriminant analysis (LDA) effect size (LEfSe) and colored by their most abundant (**A**) sites and (**B**) locations in the rat GIT. The dot size is proportional to the global relative abundance of the corresponding module. In panel (**A**), the whole GI axis was involved in biomarker mining, whereas in panel (**B**), only the adjacent mucosal and luminal compartments from the large intestine were involved. Around the cladogram are located 8 heatmap rings, where the saturation of each cell represents the average relative abundance of each module in different sampling sites (from within outward: caecal contents; proximal-, middle-, distal-colonic contents; caecal mucus; and proximal-, middle- and distal-colonic mucus). All relative abundances are z-score normalized in the radial direction for visualization. The complete lists of modules are presented in Appendix A.

**Figure 3 ijms-20-01232-f003:**
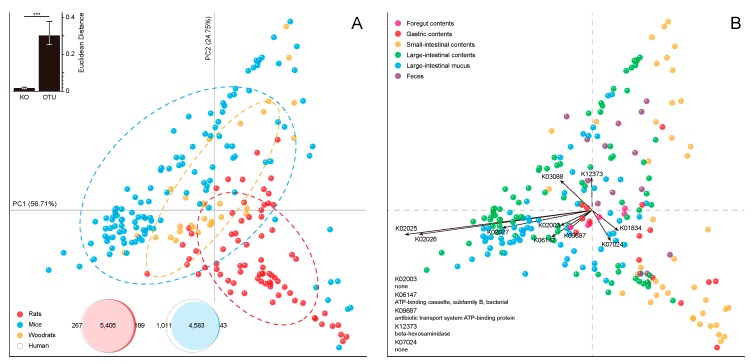
The similarities and differences in metabolic potentials among the GI microbiomes from different hosts: the PCA based on the KO profiles of GI microbiome used 227 samples form mice, rats and woodrats, grouped by (**A**) host genotype (adonis: *R*^2^ = 0.27; *p* ≤ 0.001) and (**B**) anatomic region (*R*^2^ = 0.09; *p* ≤ 0.001) in different colorations. The percentages of variation explained by PC1 and PC2 are listed along the axes representing them. The top 10 KOs driving the difference are indicated in panel (**B**). In panel (**A**), the normal confidence ellipses cover 67% of the samples belonging to the corresponding hosts. The insert graph on the upper left summarizes the Euclidean distances between samples from the same region but different hosts, calculated from KO profiles and OTU abundances respectively. The bar height represents the median inter-sample distance and the error bars range from the 25th to 75th quantile. The insert graph on the lower left shows the Venn diagrams demonstrating the functional overlaps between different mammalian hosts in the large-intestinal mucosal microbiome.

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
