# Peer review of "Potential Functions of the Gastrointestinal Microbiome Inhabiting the Length of the Rat Digest Tract"

_ijms, 2019, doi:10.3390/ijms20051232_

Reviewer 1 Report

1. The title for the manuscript is too vague, it should be more specific. 

2. There is too much unnecessary information in the introduction section. The general history of microbial metagenomic research (line 32-45) can be removed and include more information that is relevant to this research. 

3. Line 252-255: please include the number of raw read samples per project, retrieved for this research. 

Author Response

Response to Reviewer 1 Comments

Point 1: The title for the manuscript is too vague, it should be more specific.

Response 1: Thank you for your kind advice. We have changed the title from “Potential Functions of the Rat Gastrointestinal Microbiome” to “Potential Functions of the Gastrointestinal Microbiome Inhabiting along the Length of the Rat Digest Tract”.

Point 2: There is too much unnecessary information in the introduction section. The general history of microbial metagenomic research (line 32-45) can be removed and include more information that is relevant to this research.

Response 2: Thank you for your kind advice. We have rewritten the introduction section as you suggest in the revised manuscript at line 33-51.

Point 3: Line 252-255: please include the number of raw read samples per project, retrieved for this research.

Response 3: Thank you for your kind advice. We have included the number of raw read samples per project in the revised manuscript at line 255-257.

Reviewer 2 Report

I only suggest to split the section Results and Discussion into two sections, namely Results, Discussion 

Author Response

Response to Reviewer 2 Comments

Point 1: Excellent paper!I only suggest to split the section Results and Discussion into two sections, namely Results, Discussion.

Response 1: Thank you for your kind advice. This work mainly aims to complement the rat microbial biogeography dataset we released previously and tries to characterise the rat gastrointestinal microbiome from the perspective of functional potential. The results section included many comparisons between the rat gut microbiota structure and microbiome function and we frequently cited the article we published in 2017. In order to make our points of view clearer, we chose to put the results and discussion sections together.